# Management and Outcomes of Very Long-Chain Acyl-CoA Dehydrogenase Deficiency (VLCAD Deficiency): A Retrospective Chart Review

**DOI:** 10.3390/ijns10020029

**Published:** 2024-03-30

**Authors:** Maria Al Bandari, Laura Nagy, Vivian Cruz, Stacy Hewson, Alomgir Hossain, Michal Inbar-Feigenberg

**Affiliations:** 1Division of Clinical and Metabolic Genetics, Department of Pediatrics, The Hospital for Sick Children, Toronto, ON M5G 1X8, Canada; vivian.cruz@sickkids.ca; 2Division of Clinical and Metabolic Genetics, Department of Clinical Dietetics, The Hospital for Sick Children, Toronto, ON M5G 1X8, Canada; laura.nagy@sickkids.ca; 3Division of Clinical and Metabolic Genetics, Lawrence S, Bloomberg, Faculty of Nursing, University of Toronto, Toronto, ON M5T 1P8, Canada; 4Department of Genetic Counselling, The Hospital for Sick Children, Toronto, ON M5G 1X8, Canada; stacy.hewson@sickkids.ca; 5Department of Molecular Genetics, University of Toronto, Toronto, ON M5S 1A1, Canada; 6Clinical Research Services (CRS), The Hospital for Sick Children, Toronto, ON M5G 1X8, Canada; mdalomgir.hossain@sickkids.ca; 7Department of Pediatrics, University of Toronto, Toronto, ON M5S 1A1, Canada

**Keywords:** very long-chain acyl-CoA dehydrogenase deficiency, residual enzyme activity, newborn screening

## Abstract

Very long-chain acyl-CoA dehydrogenase (VLCAD) deficiency is a rare genetic condition affecting the mitochondrial beta-oxidation of long-chain fatty acids. This study reports on the clinical outcomes of patients diagnosed by newborn screening with VLCAD deficiency comparing metabolic parameters, enzyme activities, molecular results, and clinical management. It is a single-center retrospective chart review of VLCAD deficiency patients who met the inclusion criteria between January 2002 and February 2020. The study included 12 patients, 7 of whom had an enzyme activity of more than 10%, and 5 patients had an enzyme activity of less than 10%. The Pearson correlation between enzyme activity and the C14:1 level at newborn screening showed a *p*-value of 0.0003, and the correlation between enzyme activity and the C14:1 level at diagnosis had a *p*-value of 0.0295. There was no clear correlation between the number of documented admissions and the enzyme activity level. Patients who had a high C14:1 value at diagnosis were started on a diet with a lower percentage of energy from long-chain triglycerides. The C14:1 result at diagnosis is the value that has been guiding our initial clinical management in asymptomatic diagnosed newborns. However, the newborn screening C14:1 value is the most sensitive predictor of low enzyme activity and may help guide dietary management.

## 1. Introduction

Very long-chain acyl-CoA dehydrogenase (VLCAD) deficiency is a rare genetic condition affecting the mitochondrial beta-oxidation of long-chain fatty acids. It is the second most common disorder of fatty acid oxidation in Europe and the USA. The prevalence of VLCAD deficiency is estimated at 1:30,000–1/100,000 births [1]. VLCAD is associated with a range of clinical phenotypes, including a severe form with early-onset cardiac abnormalities, a hepatic or hypoketotic hypoglycemia form, and a later-onset myopathic form with intermittent rhabdomyolysis [2]. The myopathic type is likely the most common type of VLCAD deficiency [3].

A genotype–phenotype correlation was reported for VLCAD deficiency [2]. Severe disease is associated with little/no residual enzyme activity and often results from homozygosity or compound heterozygosity for null variants. Milder forms are often associated with residual enzyme activity. However, this correlation is far from perfect. The effect of novel acyl-CoA dehydrogenase very long-chain (ACADVL) variants identified in pre-symptomatic patients diagnosed through newborn screening is less clear, and the same is true for compound heterozygosity variants. For these reasons, functional assays like VLCAD enzyme activity are essential to properly differentiate carriers from mildly affected individuals and to understand the severity of novel variants [4].

Schiff et al. (2013) reported their results assessing enzyme activity on fibroblasts of 13 patients suspected to have VLCAD deficiency, but the diagnosis was uncertain. They confirmed that VLCAD activity was abnormal in nine patients. The authors concluded that functional testing adds value to differentiate affected individuals from carrier status, especially since the molecular heterogeneity of VLCAD makes it difficult to predict the functional effects based on the genotype alone [5]. However, performing enzyme activity assays on fibroblasts is invasive, time-consuming, and less practical when a clinician aims to confirm or rule out a diagnosis in the context of NBS.

The management of VLCAD deficiency varies along the spectrum of disease. Avoiding prolonged fasting and illness precautions are commonly recommended, as well as dietary management. The dietary management of VLCAD deficiency involves restricting long-chain triglycerides (LCT) and supplementing the diet with medium-chain triglycerides (MCT), which do not require the long-chain acyl-CoA dehydrogenase enzyme for energy production. All patients require periodic clinical surveillance, including physical examination and cardiac monitoring [2,4], and periodic surveillance for biochemical markers [6].

Newborn screening (NBS) for VLCAD deficiency began in Ontario in 2006. The acylcarnitine profile analysis is performed on dry blood spot (DBS) samples using MS/MS, and the main marker measured is C14:1 (https://www.newbornscreening.on.ca/, accessed on 24 March 2024). The diagnostic protocol for screen-positive patients involves the confirmatory measurement of the acylcarnitine profile in the blood (plasma or serum) and VLCAD enzyme activity analysis from lymphocytes and, if abnormal, the molecular testing of the ACADVL gene. The protocol used in Ontario differs from other screening protocols in the US and Australia, where researchers have published their outcomes for patients with VLCAD deficiency who are identified through NBS [7,8], as the majority of their patients do not have their enzyme activity measured.

Performing NBS for VLCAD deficiency led to the detection of an increased number of patients with milder phenotypes. For these patients, careful management without overtreatment is required [3].

Our primary objective was to report on the clinical outcomes of patients diagnosed with VLCAD deficiency (measured as the number of hospital admissions) comparing metabolic parameters, enzyme activities, molecular results, and dietary management. Our secondary objective was to correlate clinical outcomes and treatment with enzyme activity and molecular results.

## 2. Materials and Methods

This study was approved by the Research Ethics Board at The Hospital for Sick Children. This single-center retrospective chart review included 12 VLCAD patients under the age of 18 years followed in our metabolic clinic between January 2002 and February 2020. The Hospital for Sick Children is a tertiary referral center. All patients/guardians signed informed consent prior to participation in this study.

The inclusion criteria included patients (males or females) ≤ 18 years of age and followed by our department at The Hospital for Sick Children and those who had a confirmed diagnosis of VLCAD deficiency. The exclusion criteria included patients with abnormal acylcarnitine profile results but not confirmed to have VLCAD, patients who did not have enzyme activity results, and those with false-positive newborn screening. Out of the 17 eligible patients, 12 patients signed informed consent and were included in the study.

Data collected from the electronic record included current age and age at diagnosis, newborn screening results, confirmatory acylcarnitine profile results at diagnosis, total and free carnitine, creatine kinase (CK) including the highest CK level, liver transaminases, lactate, blood gas, urine organic acid, residual VLCAD enzyme activity, molecular genetic results, echocardiography, electrocardiogram (ECG), number of documented admissions, and dietary information including energy percentage from MCT and LCT at diagnosis and the most recent values. For patients reviewed in this retrospective study, we sent samples to Universitats Klinikum Freiberg, Freiberg, Germany, for the analysis of the VLCAD enzyme activity and the lab assessed palmitoyl-CoA oxidation (mU/mg Protein) %.

Data were collected in the Excel program, and Pearson’s correlation analysis was implemented. Categorical variables are presented as frequency and percentages, while continuous variables are presented as mean with standard deviation. Pearson’s correlation analysis was conducted between variables to obtain correlation coefficients including *p*-values to determine association. Statistical analyses were conducted using SAS (version 9.4 SAS Institute Inc., Cary, NC, USA).

## 3. Results

This study included twelve patients, and of those, seven patients had enzyme activity higher than 10% and hence were predicted to have milder disease, and five patients had enzyme activity less than 10% and hence were predicted to have a more severe presentation [2]. Eleven patients were diagnosed with NBS, and one patient was diagnosed prenatally. This patient had a sibling diagnosed with VLCAD, which prompted the prenatal diagnosis. The patient had an enzyme activity of 1%. Table 1 outlines the demographic of the patients. We reviewed clinical data including neurological symptoms (lethargy, hypotonia, and encephalopathy), liver involvement, and cardiac involvement. We did not review the development in this study. One patient had an episode of brief resolved unexplained event (BRUE) at diagnosis. It was later considered associated with GI reflux and not VLCAD deficiency. This patient had an enzyme activity of 24%. One patient had a right aortic arch with an aberrant left subclavian artery on cardiac echocardiogram, which was an incidental finding not associated with VLCAD deficiency. None of the patients had an initial presentation with metabolic decompensation at diagnosis, and their growth continues to be appropriate.

In regard to the correlation between the enzyme activity and the C14:1 value on DBS samples using MS/MS in the newborn screening for these twelve patients, a higher C14:1 level was noted in patients with lower enzyme activity. The same was seen with the C14:1 level determined at diagnosis using MS/MS on plasma/serum (Table 2). 

With respect to the number of documented admissions in relation to enzyme activity, no clear correlation was found. There were high numbers of documented illness episodes leading to emergency department visits even in patients who had higher enzyme activity and were suspected to be milder. For example, the patient who had an enzyme activity of 20% had 10 documented illnesses and 7 hospital admissions. When looking at the number of hospital admissions, overall, patients with lower enzyme activity had more admissions, and these hospital admissions were admissions to a hospital ward due to an illness that resulted in metabolic decompensation or an admission to prevent decompensation (Table 3). Pearson’s correlation between enzyme activity and the number of admissions had a *p*-value of 0.4957. Pearson’s correlation between enzyme activity and the C14:1 level determined as a part of the newborn screening was significant, with a *p*-value of 0.0005. The correlation between enzyme activity and the C14:1 level determined at diagnosis had a *p*-value of 0.0358 (Table 4).

There was no correlation between the enzyme activity and CK levels at diagnosis, nor the highest documented CK value. Pearson’s correlation between enzyme activity and the CK value at diagnosis had a *p*-value of 0.1526, and the correlation between enzyme activity and the highest CK level showed a *p*-value of 0.1771.

Table 5 shows the molecular results of the ACADVL gene and the enzyme activity for each patient. The diagnosis of VLCAD deficiency was initially confirmed based on genetic testing in amniocentesis. Out of the 12 patients, there were 5 patients with the c.848 T>C, p. (Val283Ala) molecular variant. Of those, four patients were found to be compound heterozygous with a second pathogenic variant, and one patient was homozygous. This variant is frequently reported in patients diagnosed with VLCAD deficiency by NBS and is considered associated with a milder phenotype [7,9].

The relationship between dietary management and enzyme activity was explored. There was a significant correlation between enzyme activity and the percentage of energy from MCTand LCT for both the dietary management that was started at diagnosis and for current dietary management. Patients with lower enzyme activity had a significantly higher percentage of energy in the diet provided with MCT and a lower percentage of energy provided by LCT (Table 6). Interestingly, there was no significant correlation between dietary fat provision and the C14:1 level at diagnosis, though the data suggest that a lower percentage of energy from LCT and a higher percentage of energy from MCT in the diet at the most recent assessment were associated with a higher C14:1 at diagnosis (Table 7).

## 4. Discussion

We analyzed longitudinal retrospective data of 12 patients diagnosed and treated in our metabolic center. Since VLCAD activity is regularly assessed in Ontario as a part of the confirmatory workup for NBS-positive patients, we aimed to describe its correlation to metabolic parameters’ levels, including C14:1 and CK levels, dietary management, and frequency of illnesses and admissions (as a measurement of decompensation episodes). We also report the molecular results found in our patients.

Hesse et al. (2018) reported a correlation with the initial C14:1 elevation only in patients with a residual activity ≤10% [12]. Interestingly, our results showed that there was a significant correlation between enzyme activity and the C14:1 level determined at the newborn screening (using DBS samples), with a *p*-value of 0.0005. Of note, 7 out of 12 patients had enzyme activity >10%. The correlation of enzyme activity and the C14:1 level determined at the confirmation of diagnosis (using serum or plasma) had a *p*-value of 0.0358.

The practice in our clinic is to initiate dietary management after confirmatory acylcarnitine testing demonstrates a true-positive diagnosis. Published dietary recommendations depend on the severity of VLCAD deficiency, with a higher percentage of energy from MCTs initiated in the diet of patients when severe–moderate disease is suspected. Mild patients may not be treated with MCT-based formulas or long-chain fat restrictions and will be treated by avoiding prolonged fasting and the use of a sick day management protocol as per current guidelines [6]. Disease severity is ascertained initially in our clinic through biochemical testing and clinical presentation at newborn screening retrieval and later through enzyme activity and genetic testing. With this practice in mind, it is interesting that dietary therapy initiated at diagnosis was not significantly correlated with the C14:1 level at diagnosis but was associated with enzyme activity, though enzyme activity is usually not available at the time to diet is started. There are a few possible explanations for this result. Most of the patients that had C14:1 > 1 μmol/L on diagnosis were treated with dietary modifications. Patient number 7 (later confirmed to have an enzyme activity of 14%) had a high C14:1 level on diagnosis but did not receive dietary treatment. Patient number 3 (later confirmed to have a residual enzyme activity of 3%) had a C14:1 level below 1 mmol/L; however, they received dietary treatment. The clinical team is likely using other clinical parameters at diagnosis to determine the newborn’s initial dietary therapy that are not captured in this small study and may include a clinical exam, consideration of both the NBS and confirmatory C14:1 level on retrieval, the overall acylcarnitine profile, the newborn’s feeding, and parental/physician comfort with the newborn’s diagnosis and management. Ultimately, if our decision regarding dietary treatment at diagnosis is later associated with enzyme activity, we are making good assumptions in the neonatal period about disease severity, likely from a combination of factors in addition to the C14:1 level. Another explanation is that variations like the ones described for patients numbers 3 and 7 had a significant effect on the results in this small cohort of 12 patients.

Our data did not show a significant correlation between enzyme activity and CK values at diagnosis or the highest documented CK during admission or an illness episode. Of note, CK values at diagnosis were taken in all positive newborn screening patients as a part of our initial workup protocol, even if the patients did not have an acute illness when diagnosed. The lack of significant correlation between the highest CK level and enzyme activity could be explained by our small cohort where most of the patients had a milder presentation. Only one patient had a significantly high CK level at 76,656 U/L. Other levels of highest CK were either normal or mildly elevated and ranged between 56 and 3764 U/L.

No significant correlation was found between the number of documented admissions and the enzyme activity level. It is important to note that other factors, not related to disease severity or metabolic decompensation, could affect the number of admissions. For example, parental anxiety and caregivers’ level of comfort in treating the child at home using a sick day management protocol could have also contributed to an increased number of emergency room visits and admissions.

The genotype–phenotype correlation in VLCAD suggests that those with severe mutations that result in little/no residual enzyme activity will have a more severe disease [4]. Given the association demonstrated in this study between the C14:1 value from newborn screening and enzyme activity, it may be helpful for the medical team to use the newborn screening C14:1 level as a predictor of disease severity to guide dietary management.

In regard to molecular genetic results, the c.848 T>C, p. (Val283Ala) pathogenic variant was found in five of our patients. This is consistent with other studies that have identified this variant as the most frequent variant found in positive newborn screening cases, as at least one copy is found in ∼10% of all individuals with a positive NBS. Individuals that were homozygous for p. (Val283Ala) were reported to have a relatively mild presentation and compound heterozygote patients were reported to have both severe and mild presentations, likely affected by the second variant found [9]. The patient reported in our cohort with homozygosity for p. (Val283Ala) had an enzyme activity of 8% and clinically was asymptomatic during initial presentation, as he was detected through a positive newborn screening and then he continued to be asymptomatic. This presentation corresponds to what is reported in the literature [9]. The majority of pathogenic variants found are missense variants. The variant c.1844G>A, p.(Arg615Gln) found in patient number 10 was reclassified as likely benign. This emphasizes the importance of performing an enzyme activity assay to confirm VLCAD deficiency diagnosis. Pena et al. (2016) reported that 2 out of 46 patients with one mutation were identified [7]. A second variant may lie within promoter or intronic regions and thus not identified by testing.

Due to a small cohort of 12 patients, we were not able to correlate genetic variants and residual enzyme activity results. Evans et al. (2016) concluded that mutation analysis could be considered a predictive test but often has incomplete predictive value for the need for treatment and that long-term follow-up into adulthood may be required to further support a genotype–phenotype correlation [8]. Furthermore, Merinero et al. (2018) recommended performing enzyme analysis for all newborns presenting with mildly elevated plasma C14:1 levels ranging from 0.36 to 1 mmol/L to classify them as true cases or carriers, which would further inform treatment and clinical as well as biochemical monitoring [16].

All the reported cases in our study had a positive newborn screening for VLCAD deficiency, and in one of the cases, the patient also had a prenatal diagnosis. However, there is published evidence of rare false-negative newborn screening cases for VLCAD deficiency [17,18,19]. Spiekerkoetter et al. (2012) reported a term male infant who had initially a mildly elevated C14:1 of 0.45 μmol/L on newborn screening (the cut-off level of the screening laboratory was 0.36 μmol/L). A repeat acylcarnitine screening on day 8 of life revealed a normal acylcarnitine profile; thus, VLCAD deficiency was excluded. This patient was presented at 16 months of age with metabolic decompensation. The acylcarnitine profile showed a mildly elevated C14:1 level of 0.27, and later, the patient passed away [18]. One of the described reasons for a false-negative VLCAD deficiency newborn screening as reported by Sahai et al. (2011) is that 10% dextrose given intravenously during the first day of life might suppress fatty acid oxidation and lead to a false-negative newborn screening for VLCAD deficiency [19]. Estrella et al. (2014) mentioned that the re-evaluation and revision of protocols are important. In addition, false-negative cases were reported for other conditions screened by newborn screening programs. The authors noted that although population-based screening programs are designed to identify pre-symptomatic manifestations, there is always a balance between identifying all cases of a disorder, and the problem of keeping the false-positive results to a minimum [17].

This study’s limitations are mainly the small cohort of only 12 patients, which made the analysis of dividing them into groups of milder disease with a residual enzyme activity of >10% and severe disease with a residual enzyme activity of <10% difficult and not informative. Future studies could include other centers across the province or the country to reach conclusions regarding the correlation between molecular and enzyme activity and the impact of the enzyme activity level on treatment decisions.

## 5. Conclusions

The newborn screening C14:1 level is the most sensitive predictor of low enzyme activity and hence of disease severity and may be the best marker to guide decisions about the management of newly diagnosed asymptomatic newborns.

## Figures and Tables

**Table 1 IJNS-10-00029-t001:** Demographic information.

Patient Number	Sex	% Residual Activity	Age at Diagnosis	Age at the Time of the Chart Review	Patient Ethnicity	Consanguinity
1	M	1	prenatal	3 years	Irish/Scottish/Dutch/Canadian	No
2	M	2	5 days	2 years,	Pakistani	Yes
3	F	3	3 weeks	4 years	Italian	No
4	F	6	6 weeks	9 years	French/Scottish/Irish	No
5	M	8	1 week	5 months	Irish/Scottish/French Canadian	No
6	F	12	4 weeks	5 years	Italian	No
7	M	14	8 weeks	11 years	Chinese	No
8	M	14	9 weeks	5 years	Columbian	No
9	M	19	3 weeks	3 years	Columbian/Spain/Venezuelan	No
10	M	20	8 weeks	5 years	Italian/Belgian	No
11	F	23	1 week	2 months	Afghanistan	No
12	F	24	2 weeks	7 months	European	No

**Table 2 IJNS-10-00029-t002:** Enzyme activity and C14:1 level at NBS and diagnosis.

Patient Number	% Residual Activity	C14:1 at NBS	C14:1 at Diagnosis
1	1	3.04	3.02
2	2	4.46	2.17
3	3	1.99	0.81
4	6	3.11	4.87
5	8	2.6	1.2
6	12	1.19	0.15
7	14	0.93	2.46
8	14	1.09	0.25
9	19	1.37	0.17
10	20	0.77	0.22
11	23	0.68	0.42
12	24	1.09	0.38

**Table 3 IJNS-10-00029-t003:** Enzyme activity, number of documented illness episodes, and number of hospital admissions.

Patient Number	% Residual Activity	Number of Documented Illness Episodes	Number of Hospital Admissions
1	1	4	3
2	2	11	5
3	3	3	1
4	6	8	3
5	8	4	1
6	12	5	1
7	14	4	0
8	14	11	3
9	19	5	0
10	20	10	7
11	23	2	0
12	24	2	1

**Table 4 IJNS-10-00029-t004:** Pearson correlation statistics for enzyme activity vs. C14:1 at diagnosis and NBS and the number of admissions.

Variable	With Variable	N *	Correlation Estimate	95% Confidence Limit		*p* Value
Enzyme activity	C14:1 level at Newborn screening	12	−0.82277	−0.9448	−0.441922	0.0005
Enzyme activity	C14:1 level at diagnosis	12	−0.60415	−0.8682	−0.018876	0.0358
Enzyme activity	Number of admissions	12	−0.22324	−0.7015	0.410657	0.4957

* (N) Number of patients included in the analysis.

**Table 5 IJNS-10-00029-t005:** Enzyme activity and molecular genetic results.

Patient Number	% Residual Activity	ACADVL Molecular Genetic Results Variants/ClinVar Classification	Variant Reference
1	1	1-c.753-2A>C, IVS9-2 A>C (likely pathogenic) 2-c.1388G>A, p.(Gly463Glu) (Conflicting classifications)	1-PMID: 9973285 [10] 2-PMID: 9973285 [10]
2	2	c.1349G>A, p.(Arg450His) (likely pathogenic) homozygous	PMID: 9973285 [10]
3	3	1-c.753-2A>C, IVS9-2 A>C, (likely pathogenic) 2-c.1700G>A, p.(Arg567Gln), (pathogenic)	1-PMID: 9973285 [10] 2-PMID: 20056241 [11]
4	6	1-c.848 T>C, p.(Val283Ala), (pathogenic) 2-c.1182+1G>A, IVS11+1G>A, (likely pathogenic)	1-PMID: 9973285 [10] 2-PMID: 27209629 [7]
5	8	c.848T>C, p. (Val283Ala), (pathogenic) homozygous	PMID: 9973285 [10]
6	12	1-c.865G>A, p.(Gly289Arg), (likely pathogenic) 2-c.1253G>A, p.(Ser418Asn), (pathogenic)	1-PMID: 27246109 [8] 2-PMID: 30194637 [12]
7	14	1-c.1405 C>T, p.(Arg469Trp) (likely pathogenic) 2-c.1435-3 T>A, (VUS)	1-PMID: 9973285 [10] 2-PMID: 17576681 [13]
8	14	1-c.848T>C, p.(Val283Ala), (pathogenic) 2-c.1500_1502del, p.(Leu502del), (previously reported as causing disease) heterozygous *	1-PMID: 9973285 [10] 2-PMID: 18414213 [14]
9	19	1-c.848 T>C, p.(Val283Ala) (pathogenic) 2-c.575 T>G, p.(Phe182Cys) (likely pathogenic) heterozygous *	1-PMID: 9973285 [10] 2-Novel
10	20	1-c.1844G>A, p.(Arg615Gln), (likely benign) 2-c.1322G>A, p.(Gly441Asp), (pathogenic)	1-PMID: 30194637 [12] 2-PMID: 30194637 [12]
11	23	1-c.848T>C, p.(Val283Ala) (pathogenic) 2-c.535G>T, p.(Gly179Trp) (VUS)	1-PMID: 9973285 [10] 2-PMID: 19327992 [15]
12	24	1-c.339T>C>A, p.(Phe113Leu) (VUS) 2-c.833_835del, p.(Lys278del) (pathogenic) *	1-PMID: 30194637 [12] 2-PMID: 30194637 [12]

* Not listed in ClinVar.

**Table 6 IJNS-10-00029-t006:** Pearson correlation statistics for enzyme activity vs. % energy intake from LCT and MCT in the diet started at diagnosis and most recent value.

Variable	With Variable	N *	Correlation Estimate	95% Confidence Limit		*p* Value
Enzyme activity	LCT% value at diagnosis	9	0.84884	0.37892	0.963964	0.0022
Enzyme activity	MCT% value at diagnosis	9	−0.84748	−0.9636	−0.374835	0.0023
Enzyme activity	Most recent LCT% value	9	0.78323	0.20175	0.947307	0.0099
Enzyme activity	Most recent MCT% value	9	−0.73475	−0.9346	−0.092633	0.0214

* (N) Number of patients included in the analysis.

**Table 7 IJNS-10-00029-t007:** Pearson correlation statistics for C14:1 at diagnosis vs. % energy intake from LCT and MCT in the diet started at diagnosis and most recent value.

Variable	With Variable	N *	Correlation Estimate	95% Confidence Limit		*p* Value
C14:1 at diagnosis	LCT% value at diagnosis	9	−0.58129	−0.8913	0.170401	0.1036
C14:1 at diagnosis	MCT% value at diagnosis	9	0.53874	−0.2274	0.878431	0.1401
C14:1 at diagnosis	Most recent LCT% value	9	−0.64285	−0.9092	0.077163	0.0616
C14:1 at diagnosis	Most recent MCT% value	9	0.62584	−0.1043	0.904347	0.072

* (N) Number of patients included in the analysis.

## Data Availability

Data from this study can be accessed by contacting the corresponding author.

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
