# Peer review of "Management and Outcomes of Very Long-Chain Acyl-CoA Dehydrogenase Deficiency (VLCAD Deficiency): A Retrospective Chart Review"

_2409-515X, 2024, doi:10.3390/ijns10020029_

Round 1

Reviewer 1 Report

Comments and Suggestions for Authors

The authors report a single center chart review study on clinical outcomes of patients diagnosed with VLCAD deficiency. A review of 12 patients’ charts revealed that VLCAD activity and C14:1 level at the time of NBS and at diagnosis showed significant correlation. Their findings also showed a significant correlation between VLCAD activity and % of energy from MCT and LCT. Should be Ok for publication once several related research works are cited in the introduction and authors discuss the question how their chart review study may advance our understanding of VLCAD diagnosis and clinical management of the disease.

1.      Abstract: Line 25; LCT%- an abbreviation should not be used  

2.      Introduction: Several published research articles on the VLCAD deficiency, enzyme levels, diagnosis and molecular and cellular pathology were published. Authors need to reference at least the following articles and discuss how the current study will improve our understanding on patients’ outcome/treatment having VLCAD deficiency.

Evans et.al. Mol Genet Metab. 2016; 118: 282-287

Merinero et. al. JIMD Rep. 2018; 39: 63-74

Yamada and Taketani J Hum Genet. 2019; 64: 73-85

Schiff et. al. Mol Genet Metab. 2013; 109: 21-27

Line 44; describe abbreviation ACADVL

3.      Material and Methods: Line 83: A brief comment about how VLCAD enzyme activity was measured. This will be helpful to compare data from other published findings.

4.      Results: Line 100: Was there anything unusual with respect to the diagnosis, biochemical or genotype information on the pre-natal diagnosed patient?

Line 136; Table 4 title is missing.

5.      Discussion: Line 174; Can authors provide a plausible explanation for the chart finding?

Line 206-207; correct typo/grammar mistake

Line 208; 2/12 should be 2 out of 12 patients.

A discussion of the Spanish and Australian follow-up and NBS diagnosis study data will be helpful. In addition, comparing the current chart review data with the information of interpreting genetic findings, and with clinical, biochemical, and enzymatic monitoring for evaluation/treatment of patients (references mentioned above) will also be very helpful for NBS community.  

Reviewer 2 Report

Comments and Suggestions for Authors

The authors collected 12 cases of VLCAD deficiency in one hospital over a 20 year period. This is an important observation to evaluate the value of newborn screening for this rare metabolic disorder. The authors demonstrate the value of C14:1 at screening is correlated with levels of residual enzyme activity and enzyme activity is correlated with patients’ diet. Because of the value of these patients, we want to see more data from these patients, besides the statistic values.

Clinic data that would enrich this paper include the demographic information of patients, like sex, current age, growth (weight, height), development (any mental impairment), etc. The authors mention that dietary managements depended on C14:1 level. We would like to see serial changes in dietary component and may be also the corresponding C14:1 level, could be on a chart. Please associate the p values to the table or chart, whenever possible.

Round 2

Reviewer 2 Report

Comments and Suggestions for Authors

no more comment
